# ABDUCTIVE KNOWLEDGE INDUCTION FROM RAW DATA

## ABSTRACT

For many reasoning-heavy tasks, it is challenging to find an appropriate end-to-end differentiable approximation to domain-specific inference mechanisms. Neural-Symbolic (NeSy) AI divides the end-to-end pipeline into neural perception and symbolic reasoning, which can directly exploit general domain knowledge such as algorithms and logic rules. However, it suffers from the exponential computational complexity caused by the interface between the two components, where the neural model lacks direct supervision, and the symbolic model lacks accurate input facts. As a result, they usually focus on learning the neural model with a sound and complete symbolic knowledge base while avoiding a crucial problem: where does the knowledge come from? In this paper, we present Abductive Meta-Interpretive Learning ($Meta_{Abd}$), which unites abduction and induction to learn perceptual neural network and first-order logic theories simultaneously from raw data. Given the same amount of domain knowledge, we demonstrate that $Meta_{Abd}$ not only outperforms the compared end-to-end models in predictive accuracy and data efficiency but also induces logic programs that can be re-used as background knowledge in subsequent learning tasks. To the best of our knowledge, $Meta_{Abd}$ is the first system that can jointly learn neural networks and recursive first-order logic theories with predicate invention.

## 1 INTRODUCTION

Inductive bias, background knowledge, is an essential component in machine learning. Despite the success of data-driven end-to-end deep learning in many traditional machine learning tasks, it has been shown that incorporating domain knowledge is still necessary for some complex learning problems (Dhingra et al., 2020; Grover et al., 2019; Trask et al., 2018).

In order to leverage complex domain knowledge that is discrete and relational, end-to-end learning systems need to represent it with a differentiable module that can be embedded in the deep learning context. For example, graph neural networks (GNN) use relational graphs as an external knowledge base (Zhou et al., 2018); some works even considers more specific domain knowledge such as differentiable primitive programs (Gaunt et al., 2017). However, the design of these modules is usually *ad hoc*. Sometimes, it is not easy to find an appropriate approximation that is suited for single-model based end-to-end learning (Glasmachers, 2017; Garcez et al., 2019).

Therefore, many researchers propose to break the end-to-end learning pipeline apart and build a hybrid model that consists of smaller modules where each of them only accounts for one specific function (Glasmachers, 2017). A representative branch in this line of research is Neural-Symbolic (NeSy) AI (De Raedt et al., 2020; Garcez et al., 2019) aiming to bridge System 1 and System 2 AI (Kahneman, 2011; Bengio, 2017), i.e., neural-network-based machine learning and symbolic-based relational inference. In NeSy models, the neural network extracts high-level symbols from noisy raw data and the symbolic model performs relational inference over the extracted symbols.

However, the non-differentiable interface between neural and symbolic systems (i.e., the facts extracted from raw data and their truth values) leads to high computational complexity in learning. For example, due to the lack of direct supervision to the neural network and reliable inputs to the symbolic model, some works have to use Markov Chain Monte Carlo (MCMC) sampling or zero-order optimisation to train the model (Li et al., 2020; Dai et al., 2019), which could be inefficient in practice. Consequently, almost all hybrid models assume the existence of a very strong predefined

domain knowledge base and focus on using it to train neural networks. It limits the expressive power of the hybrid-structured model and sacrifices many benefits of symbolic learning (e.g., predicate invention, learning recursive theories, and re-using learned models as background knowledge).

In this paper, we integrate neural networks with Inductive Logic Programming (ILP) (Muggleton & de Raedt, 1994)—a general framework for symbolic machine learning—to enable first-order logic theory induction from raw data. More specifically, we present Abductive Meta-Interpretive Learning ($Meta_{abd}$) which extends the Abductive Learning (ABL) framework (Dai et al., 2019; Zhou, 2019) by combining logical induction and abduction (Flach et al., 2000) with neural networks in Meta-Interpretive Learning (MIL) (Muggleton et al., 2014). $Meta_{Abd}$ employs neural networks to extract probabilistic logic facts from raw data, and induces an abductive logic program (Kakas et al., 1992) that can efficiently infer possible truth values of the probabilistic facts to train the neural model.

On the one hand, the abductive logic program learned by $Meta_{Abd}$ can largely prune the search space of the truth value assignments to the logical facts extracted by an under-trained neural model. On the other hand, the extracted probabilistic facts, although noisy, provide a distribution on the possible worlds (Nilsson, 1986) reflecting the raw data distribution, which helps logical induction to identify the most probable hypothesis. The two systems in $Meta_{Abd}$ are integrated by a probabilistic model that can be optimised with Expectation Maximisation (EM).

To the best of our knowledge, $Meta_{abd}$ is the first system that can simultaneously (1) train neural models, (2) learn recursive logic theories and (3) perform predicate invention from domains with sub-symbolic representation. In the experiments we compare $Meta_{Abd}$ to the compared state-of-the-art end-to-end deep learning models on two complex learning tasks. The results show that, given the same amount of background knowledge, $Meta_{abd}$ outperforms the end-to-end models significantly in terms of predictive accuracy and data efficiency, and learns human interpretable models that could be re-used in subsequent learning tasks.

## 2 RELATED WORK

Solving "System 2" problems require the ability of relational and logical reasoning instead of "intuitive and unconscious thinking" (Kahneman, 2011; Bengio, 2017). Due to the complexity of this type of tasks, many researchers have tried to embed intricate background knowledge in end-to-end deep learning models. For example, Trask et al. (2018) propose the differentiable Neural Arithmetic Logic Units (NALU) to model basic arithmetic functions (e.g., addition, multiplication, etc.) in neural cells; Grover et al. (2019) encode permutation operators with a stochastic matrix and present a continuous and differentiable approximation to the sort operation; Wang et al. (2019) introduce a differentiable SAT solver to enable gradient-based constraint solving. However, most of these specially designed differentiable modules are *ad hoc* approximations to the original inference mechanisms, which can not represent the inductive bias in a general form such as formal languages.

In order to directly exploit the complex background knowledge expressed by formal languages, Statistical Relational (StarAI) and Neural Symbolic (NeSy) AI (De Raedt et al., 2020; Garcez et al., 2019) are proposed. Some works try to approximate logical inference with continuous functions or use probabilistic logical inference to enable the end-to-end training (Cohen et al., 2020; Manhaeve et al., 2018; Donadello et al., 2017); others try to combine neural networks and pure symbolic reasoning by performing a combinatorial search over the truth values of the output facts of the neural model (Li et al., 2020; Dai et al., 2019). Because of the highly complex statistical relational inference and combinatorial search, it is difficult for them to learn first-order logic theories. Therefore, most existing StarAI and NeSy systems focus on utilising a pre-defined symbolic knowledge base to help the parameter learning of the neural model and probabilistic model.

One way to learn symbolic models is to use Inductive Logic Programming (Muggleton & de Raedt, 1994). Some early work on combining logical abduction and induction can learn logic theories even when input data is incomplete (Flach et al., 2000). Recently, $\partial$ILP was proposed for learning first-order logic theories from noisy data (Evans & Grefenstette, 2018). However, these works are designed for learning from domains. Otherwise, they need to use a fully trained neural model to extract primitive facts from raw data before symbolic learning. Machine apperception (Evans et al., 2019) unifies reasoning and perception by combining logical inference and binary neural networks in Answer Set Programming, in which logic hypotheses and parameters of neural networks are all represented by logical groundings, making the system hard to optimise. For problems involving noisy inputs like MNIST images, it still requires a fully pre-trained neural net for pre-processing.

Different to the previous work, our presented Abductive Meta-Interpretive Learning ($Meta_{Abd}$) aims to combine symbolic and sub-symbolic learning in a mutually beneficial way, where the induced abductive logic program prunes the combinatorial search of the unknown labels for training the neural model; and the probabilistic facts output by the neural model provide a distribution on the possible worlds of the symbolic domain to help logic theory induction.

## 3 ABDUCTIVE META-INTERPRETIVE LEARNING

### 3.1 PROBLEM FORMULATION

A typical hybrid model bridging sub-symbolic and symbolic learning contains two major parts: a perception model and a reasoning model (Dai et al., 2019). The perception model maps raw inputs $x \in \mathcal{X}$—which are usually noisy and represented by sub-symbolic features—to some primitive symbols $z \in \mathcal{Z}$, such as digits, objects, ground logical expressions, etc. The reasoning model takes the interpreted $z$ as input and deduces the final output $y \in \mathcal{Y}$ according to a symbolic background knowledge base $B$. Because the primitive symbols $z$ are *uncertain* and *not observable* from both training data and the background knowledge, we have named them as *pseudo-labels* of $x$.

The perception model is parameterised with $\theta$ and outputs the conditional probability $P_\theta(z|x) = P(z|x,\theta)$; the reasoning model $H \in \mathcal{H}$ is a set of *first-order* logical clauses such that $B \cup H \cup z \models y$, where "$\models$" means "logically entails". Our target is to learn $\theta$ and $H$ simultaneously from training data $D = \{\langle x_i, y_i \rangle\}_{i=1}^n$. For example, if we have one example with $x = $ [1, 2, 3] and $y = 6$, given background knowledge about adding two numbers, the hybrid model should learn a perception model that recognises $z = [1, 2, 3]$ and induce a program to add all numbers in $z$ recursively.

Assuming that $D$ is an i.i.d. sample from the underlying distribution of $(x, y)$, the objective of our learning problem can be represented as follows:

$$(H^*, \theta^*) = \arg\max_{H, \theta} \prod_{\langle x, y \rangle \in D} \sum_{z \in \mathcal{Z}} P(y, z | B, x, H, \theta), \tag{1}$$

where $z$ is a hidden variable in this model. Theoretically, this problem can be solved by Expectation Maximisation (EM) algorithm. However, even if we can obtain the expectation of the hidden variable $z$ and efficiently estimate the perception model's parameter $\theta$ with numerical optimisation, the hypothesis $H$, which is a first-order logic theory, is still difficult to be optimised together with $\theta$.

We propose to solve this problem by treating $H$ like $z$ as an extra hidden variable, which gives us:

$$\theta^* = \arg\max_{\theta} \prod_{\langle x, y \rangle \in D} \sum_{H \in \mathcal{H}} \sum_{z \in \mathcal{Z}} P(y, H, z | B, x, \theta) \tag{2}$$

The hybrid-model learning problem in Equation 1 can be split into two EM steps: (1) **Expectation:** obtain the expected value of $H$ and $z$ by sampling them in their discrete hypothesis space from $(H, z) \sim P(H, z | B, x, y, \theta)$; (2) **Maximisation:** estimate $\theta$ by maximising the likelihood of training data with efficient numerical optimisation approaches such as gradient descent.

As one can imagine, the main challenge is to estimate the expectation of the hidden variables $H \cup z$, i.e., we need to search for the most probable $H$ and $z$ given the $\theta$ learned in the previous iteration.

**Challenges** This search problem is nontrivial. Sampling the values of hidden variable $z$ results in a search space growing exponentially with the number of training examples. Even when $B$ is sound and complete, existing hybrid models that do not learn first-order hypotheses still have to use Zero-Order Optimisation (ZOOpt) or Markov Chain Monte Carlo (MCMC) sampling to estimate the expectation of $z$ (Dai et al., 2019; Li et al., 2020), which could be quite inefficient in practice.

Furthermore, the size and structure of hypothesis space $\mathcal{H}$ of first-order logic programs makes the search problem even more complicated. For example, given $x = $ [1, 2, 3] and $y = 6$, when the perception model is accurate enough to output the most probable $z = [1, 2, 3]$, we have at least two choices for $H$: cumulative sum or cumulative product. When the perception model is under-trained and outputs the most probable $z = [2, 2, 3]$, then $H$ could be a program that only multiplies the last two digits. Hence, $H$ and $z$ are entangled and cannot be treated independently.

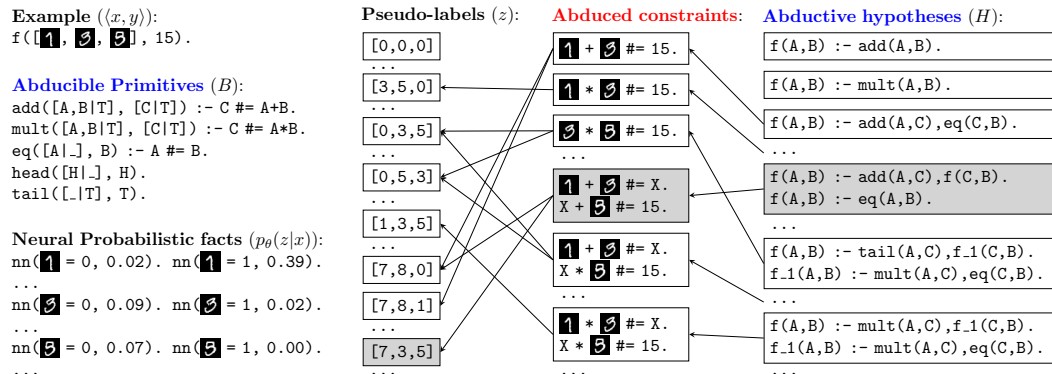

Figure 1: Example of $Meta_{Abd}$'s abduction-induction learning. Given training examples, background knowledge of abducible primitives and probabilistic facts generated by a perceptual neural net, $Meta_{Abd}$ learns an abductive logic program $H$ and abduces relational constraints (implemented with the CLP(Z) predicate "#="[1]) over the input images; it then uses them to efficiently prune the search space of the most probable pseudo-labels $z$ (in grey blocks) for training the neural network.

## 3.2 EFFICIENT HYPOTHESIS SAMPLING BY COMBINING ABDUCTION AND INDUCTION

Inspired by early works in abductive logic programming (Flach et al., 2000), we propose to solve the challenges above by combining logical induction and abduction. The induction learns an abductive logic theory $H$ based on $P_\theta(z|x)$, and the abduction by $H$ reduces the search space of $z$.

*Abduction* refers to the process of selectively inferring specific grounded facts and hypotheses that give the best explanation to observations based on background knowledge of a deductive theory. For example, if we know that $H$ is a cumulative sum program and observe that $x = $ ▣, ▣, ▣ and $y = 6$, then we can *abduce* that $x$ must satisfy the constraint Z1+Z2+Z3=6, where $[Z1, Z2, Z3] = z$ are the pseudo-labels of images in $x$. This constraint can largely prune the search space of $z$, in which all Zi > 6 can be excluded. If the current perception model assigns very high probabilities to Z1 = 2 and Z2 = 3, one can easily infer that Z3 = 1 even when the perception model has relatively low confidence about it, as this is the only solution that satisfies the constraint.

An illustrative example of combining abduction and induction is shown in Figure 1. Briefly speaking, instead of directly sampling pseudo-labels $z$ and $H$ together from the huge hypothesis space, our proposed Abductive Meta-Interpretive Learning approach only samples the abductive logic program $H$, and then use the abduced relational constraints to prune the search space of $z$. Meanwhile, the perception model outputs the likelihood of pseudo-labels with $p_\theta(z|x)$ which defines a distribution over all possible values of $z$ and helps to find the most probable $H \cup z$.

Formally, for each example $\langle x, y \rangle$, we re-write the likelihood in Equation 2 as follows:

$$
\begin{aligned}
P(y, H, z | B, x, \theta) &= P(y, H | B, z) P_\theta(z|x) \\
&= P(y | B, H, z) P(H | B, z) P_\theta(z|x) \\
&= P(y | B, H, z) P_{\sigma^*}(H | B) P_\theta(z|x),
\end{aligned}
\tag{3}
$$

where $P_{\sigma^*}(H|B)$ is the Bayesian prior distribution on first-order logic hypotheses, which is defined by the transitive closure of *stochastic refinements* $\sigma^*$ given the background knowledge $B$ (Muggleton et al., 2013), where a refinement is a unit modification (e.g., adding/removing a clause or literal) to a logic theory. The equations hold because: (1) pseudo-label $z$ is conditioned on $x$ and $\theta$ since it is the output of the perception model; (2) $H$ follows the prior distribution so it only depends on $B$; (3) $y \cup H$ is independent from $x$ given $z$ because the relations among $B$, $H$, $y$ and $z$ are determined by pure logical inference, where:

$$
P(y | B, H, z) = \begin{cases} 1, & \text{if } B \cup H \cup z \vdash y, \\ 0, & \text{otherwise.} \end{cases}
\tag{4}
$$

---

[1]CLP(Z) is a constraint logic programming package accessible at https://github.com/triska/clpz. More implementation details please refer to the Appendix.

```
                    Abductive Meta-Interpreter
prove([], Prog, Prog, [], Prob, Prob).
prove([Atom|As], Prog1, Prog1, Abds, Prob1, Prob2) :-
    deduce(Atom),
    prove(As, Prog1, Prog2, Abds, Prob1, Prob2).
prove([Atom|As], Prog1, Prog1, Abds, Prob1, Prob2) :-
    call_abducible(Atom, Abd, Prob),
    Prob3 is Prob1 * Prob,
    get_max_prob(Max), Prob3 > Max,
    set_max_prob(Prob3),
    prove(As, Prog1, Prog1, [Abd|Abds], Prob3, Prob2).
prove([Atom|As], Prog1, Prog2, Abds, Prob1, Prob2) :-
    meta-rule(Name, MetaSub,(Atom :- Body), Order),
    Order,
    substitue(metasub(Name, MetaSub), Prog1, Prog3),
    prove(Body, Prog3, Prog4),
    prove(As, Prog4, Prog2, Abds, Prob1, Prob2)
```

Figure 2: Prolog code for $Meta_{Abd}$. It recursively proves a series of atomic goals in three ways: (1) deducing them from background knowledge; (2) abducing a possible grounded expression (e.g., relational constraint) to satisfy them (bold fonts); (3) matching them against the heads of meta-rules and form an augmented program or prove it with the current program. Finally, the abduced groundings Abd are used for searching the best pseudo-labels $z$; the probability of Abd is used for calculating the score function in Equation 5.

Following Bayes' rule we have $P(H, z|B, x, y, \theta) \propto P(y, H, z|B, x, \theta)$. Therefore, we can sample the most probable $H \cup z$ in the expectation step according to Equation 3 as follows:

1. Sample an *abductive* first-order logic hypothesis $H \sim P_{\sigma^*}(H|B)$;

2. Use $H \cup B$ and $y$ to *abduce*[2] possible pseudo-labels $z$, which are *guaranteed* to satisfy $H \cup B \cup z \vdash y$ and exclude the values of $z$ such that $P(y|B, H, z) = 0$;

3. According to Equation 3 and 4, tor each sampled $H \cup z$ calculate its score by:

$$score(H, z) = P_{\sigma^*}(H|B)P_\theta(z|x) \tag{5}$$

4. Return the $H \cup z$ with the highest score to continue the maximisation step.

By learning an abductive logic theory $H$, the search space of pseudo-label $z$ can be largely pruned thanks to the sparsity of the probabilistic distribution structured by $B \cup H \cup z \vdash y$.

## 3.3 THE $Meta_{abd}$ IMPLEMENTATION

We implement the above abduction-induction algorithm with Abductive Meta-Interpretive Learning ($Meta_{Abd}$), whose codes are shown in Figure 2. It extends the general meta-interpreter of MIL (Muggleton et al., 2014) by including an abduction procedure (bold fonts in Figure 2) that can abduce relational constraints on pseudo-labels $z$ for pruning the search space.

Meta-Interpretive Learning (MIL) is a form of ILP (Muggleton & de Raedt, 1994). It learns first-order logic programs with a second-order meta-interpreter, which is composed of a definite first-order background knowledge $B$ and meta-rules $M$. $B$ contains the primitive predicates for constructing first-order hypotheses $H$; $M$ is second-order clauses with existentially quantified predicate variables and universally quantified first-order variables that shape the structure of the hypothesis space $\mathcal{H}$. Briefly speaking, MIL attempts to prove the training examples and saves the resulting programs for successful proofs. However, MIL can only learn first-order logic programs from pure symbolic domains, where the examples are deterministic and noise-free.

By combining abduction and induction, $Meta_{Abd}$ can learn abductive logic programs from noisy domains where the distribution on possible worlds (Nilsson, 1986) is given by a set of probabilistic facts. A *possible world* is a truth value assignment to the probabilistic logic facts. For the example in Figure 1, each combination of the possible pseudo-labels of the three input images forms a

---

[2]The abduction can be naturally accelerated by parallel computing, more details are in the Appendix.

Table 1: Domain knowledge used by the compared models.

| Domain Knowledge | End-to-end Models | $Meta_{Abd}$ |
|---|---|---|
| Recurrence | LSTM & RNN | Prolog's list operations |
| Arithmetic functions | `NAC` & `NALU` (Trask et al., 2018) | Predicates `add`, `mult` and `eq` |
| Permutation | Permutation matrix $P_{\texttt{sort}}$ (Grover et al., 2019) | Prolog's `permutation` |
| Sorting | `sort` operator (Grover et al., 2019) | Predicate `s` (learned from sub-task) |

possible world, whose probability distribution is defined by the probability values output by the perceptual neural net. As shown in the figure, given the abducible primitives as background knowledge $Meta_{Abd}$ can construct the hypotheses $H$ while abducing the relational constraints on $z$.

After an abductive hypothesis $H$ has been sampled, the search for $z$ will be done by logical abduction. Finally, the score of $H \cup z$ will be calculated by Equation 5, where $P_\theta(z|x)$ is the output of the perception model, which in this work is implemented with a neural network $\varphi_\theta$ that outputs:

$$P_\theta(z|x) = softmax(\varphi_\theta(x, z)).$$

Meanwhile, we define the prior distribution on $H$ by following Hocquette & Muggleton (2018):

$$P_{\sigma^*}(H|B) = \frac{6}{(\pi \cdot c(H))^2},$$

where $C(H)$ is the complexity of the learned program, e.g., the size of $H$.

## 4 EXPERIMENTS

This section describes the experiments which apply $Meta_{Abd}$ to learn first-order logic programs from images of handwritten digits in two scenarios: (1) cumulative sum/product and (2) sorting. The experiments aim to address the following two questions:

1. Can the abduction-induction strategy of $Meta_{Abd}$ learn first-order logic programs and train perceptual neural networks jointly?

2. Given the same type and amount of background knowledge shown in Table 1, is hybrid modelling, which directly leverages the background knowledge in symbolic form, better than end-to-end learning?

### 4.1 LEARNING CUMULATIVE SUM AND PRODUCT FROM IMAGES

**Materials** Following the setting of Trask et al. (2018), the inputs of the two tasks are series of randomly chosen MNIST digits; the numerical outputs are the sum and product of the digits, respectively. The lengths of training sequences are 2-5. To verify if the learned models can extrapolate to longer inputs, we also include test examples with length 10 (both tasks), 15 (in the cumulative product task) and 100 (in the cumulative sum task). In the cumulative product experiments, when the randomly generated sequence is long enough, it will be very likely to contain a 0 and makes the final outputs equal to 0. So the extrapolation examples with length 15 only contain digits from 1 to 9. The dataset contains 3000 and 1000 examples for training and validation, respectively; the test data of each length has 10,000 examples. Since the end-to-end models usually require more training data due to the model complexity, we also did experiments with 10,000 training examples for them.

**Methods** We compare $Meta_{Abd}$ with four end-to-end learning baselines, including RNN, LSTM and LSTMs attached to Neural Accumulators(NAC) and Neural Arithmetic Logic Units (NALU)[3] (Trask et al., 2018). The performance is measured by classification accuracy (Acc.) on length-one inputs, mean average error (MAE) in sum tasks, and mean average error on logarithm (log MAE) of the outputs in product tasks whose error grows exponentially with sequence length.

A convnet processes the input images to the recurrent networks, as Trask et al. (2018) described; it also serves as the perception model of $Meta_{Abd}$ to output the probabilistic facts. As shown in Table 1, all models are aware of the same amount of background knowledge: the end-to-end models use LSTM or RNN to handle recurring inputs and use `NAC`s and `NALU`s to encode basic arithmetic functions, while $Meta_{Abd}$ can exploit them explicitly as primitive predicates in the Prolog language. Note that $Meta_{Abd}$ uses the same background knowledge for both sum and product tasks. Each experiment is carried out five times, and the average of the results are reported.

[3]We use the implementation of `NAC` and `NALU` from https://github.com/kevinzakka/NALU-pytorch

Table 2: Results on the MNIST cumulative sum/product tasks.

| | MNIST cumulative sum | | | | MNIST cumulative product | | | |
| --- | --- | --- | --- | --- | --- | --- | --- | --- |
| | Acc. | MAE | | | Acc. | log MAE | | |
| Seq Len | 1 | 5 | 10 | 100 | 1 | 5 | 10 | 15 |
| LSTM | 9.80% | 15.3008 | 44.3082 | 449.8304 | 9.80% | 11.1037 | 19.5594 | 21.6346 |
| RNN-Relu | 10.32% | 12.3664 | 41.4368 | 446.9737 | 9.80% | 10.7635 | 19.8029 | 21.8928 |
| LSTM-NAC | 7.02% | 6.0531 | 29.8749 | 435.4106 | 0.00% | 9.6164 | 20.9943 | 17.9787 |
| LSTM-NAC$_{10k}$ | 8.85% | 1.9013 | 21.4870 | 424.2194 | 10.50% | 9.3785 | 20.8712 | 17.2158 |
| LSTM-NALU | 0.00% | 6.2233 | 32.7772 | 438.3457 | 0.00% | 9.6154 | 20.9961 | 17.9487 |
| LSTM-NALU$_{10k}$ | 0.00% | 6.1041 | 31.2402 | 436.8040 | 0.00% | 8.9741 | 20.9966 | 18.0257 |
| $Meta_{Abd}$ | **95.27%** | **0.5100** | **1.2994** | **6.5867** | **97.73%** | **0.3340** | **0.4951** | **2.3735** |
| LSTM-NAC$_{1\text{-shot CNN}}$ | 49.83% | 0.8737 | 21.1724 | 426.0690 | 0.00% | 6.0190 | 13.4729 | 17.9787 |
| LSTM-NALU$_{1\text{-shot CNN}}$ | 0.00% | 6.0070 | 30.2110 | 435.7494 | 0.00% | 9.6176 | 20.9298 | 18.1792 |
| $Meta_{Abd+1\text{-shot CNN}}$ | 98.11% | 0.2610 | 0.6813 | 4.7090 | 97.94% | 0.3492 | 0.4920 | 2.4521 |

**Results** Our experimental results are shown in Table 2; the learned first-order logic theories are shown in Figure 3a. End-to-end models that do not exploit any background knowledge (LSTM and RNN) perform worst on these tasks. For NALU and NAC, even though they can exploit background knowledge by using the specially designed differentiable neural modules, the performance is still significantly worse than $Meta_{Abd}$ given the same amount of training data or even more.

Although $Meta_{Abd}$ achieves the best result among the compared methods, we observe that its EM learning sometimes converges to saddle points or local optima in the cumulative sum task. This phenomenon happens less in the other task, because of the distribution $P(H, z|B, x, y, \theta)$ of learning the cumulative product function is much sparser compared to cumulative sum. Therefore, we also carry out extra experiments with 1-shot pre-trained convnets, which are trained by randomly sampling one example in each class of MNIST data. Although the pre-trained convnet is weak (Acc. 20~35%), it provides a good initialisation for the EM algorithm and improves the learning performance.

**Cumulative Sum**:
```
f(A,B):-add(A,C),f(C,B).
f(A,B):-eq(A,B).
```

**Cumulative Product**:
```
f(A,B):-mult(A,C),f(C,B).
f(A,B):-eq(A,B).
```

**Bogosort**:
```
f(A,B):-permute(A,B,C),s(C).
s(A):-s_1(A,B),s(B).
s(A):-tail(A,B),empty(B).
s_1(A,B):-nn_pred(A),tail(A,B).
```

(a) Programs learned by $Meta_{Abd}$

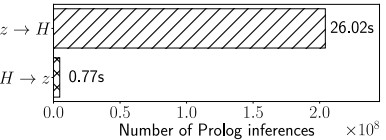

(b) Comparison of sampling $z$ and $H$

Figure 3: Learned programs and the time efficiency of abduction.

Figure 3b shows the time efficiency of $Meta_{Abd}$'s abduction-induction strategy on one batch of examples in the cumulative sum task. "$z \rightarrow H$" means first samples pseudo-labels $z$ and then learn $H$ with ILP; "$H \rightarrow z$" means first sample an abductive hypothesis $H$ and then use $H$ to abduce $z$. The unit of x-axis is the average number of Prolog inferences, the number at the end of each bar is the average inference time in seconds. Evidently, the abduction leads to a substantial improvement in the number of Prolog inferences and significantly reduces the search complexity.

## 4.2 LEARNING BOGOSORT FROM IMAGES

**Materials** Following the setting from Grover et al. (2019), the input of this task is a sequence of randomly chosen MNIST images of distinct numbers; the output is the correct ranking (from large to small) of the digits. For example, when $x = $ [5,9,4,3,8] ([5,9,4,3,8]), then the output should be $y = [3, 1, 4, 5, 2]$. The training dataset contains 3000 training and 1000 validation examples; the test dataset has 10,000 examples. The training examples contain five images, and we test the learned models on image sequences with lengths 3, 5 and 7. Experiments are repeated five times, and the average of results are reported.

**Methods** In this task, we compare $Meta_{Abd}$ to NeuralSort[4] (Grover et al., 2019), which implements a differentiable relaxation of sorting operator. Given an input list of scalars, it generates a stochastic permutation matrix by applying the pre-defined deterministic or stochastic `sort` operator on the inputs, i.e., NeuralSort can be regarded as a differentiable approximation to bogosort

---

[4]We use the implementation of NeuralSort from https://github.com/ermongroup/neuralsort

Table 3: Average accuracy of the bogosort task. First value is the rate of correct permutations; second value is the rate of correct individual element ranks.

| Seq Len | 3 | 5 | 7 |
|---|---|---|---|
| Deterministic NeuralSort | 95.49% (96.82%) | 88.26% (94.32%) | 80.51% (92.38%) |
| Stochastic NeuralSort | 95.37% (96.74%) | 87.46% (94.03%) | 78.50% (91.85%) |
| $Meta_{Abd}$ | **96.33% (97.22%)** | **91.75% (95.24%)** | **87.42% (93.58%)** |

(permutation sort). Although for $Meta_{Abd}$ it is easy to include stronger background knowledge for learning more efficient sorting algorithms like quicksort (Cropper & Muggleton, 2019), in order to make a fair comparison, we adapt the same background knowledge as NeuralSort to logic rules and learn bogosort. We did not compare to other baselines such as LSTM/RNN with different activation layers because they are weaker than NeuralSort in this task (Grover et al., 2019).

The background knowledge of permutation in $Meta_{Abd}$ is implemented with Prolog's built-in predicate `permutation`. Meanwhile, instead of providing the information about sorting as prior knowledge like the NeuralSort, we try to *learn* the concept of "sorted" (represented by a monadic predicate `s`) from data as a sub-task, whose training set is the subset of the sorted examples within the training dataset ($< 20$ examples). To do this, $Meta_{Abd}$ uses an MLP attached to the same untrained convnet as previous experiments to produce dyadic probabilistic facts `nn_pred([7, 9|_])`, which learns if the first two items in the image sequence satisfy a dyadic relation. Please note that the attached MLP is not provided with supervision on `nn_pred` about whether it should learn "greater than" or "less than". Moreover, we do *not* provide any prior knowledge about total ordering, so `nn_pred` only learns a dyadic partial order among the MNIST images. As we can see, the background knowledge used by $Meta_{Abd}$ is *much weaker* than that is used by NeuralSort. The sorting task and its sub-task are trained sequentially. In our experiments, the first five epochs of $Meta_{Abd}$ learn the sub-task, and then it re-uses the learned models to learn bogosort.

**Results**  Table 3 shows the average accuracy of the compared methods in the sorting tasks; Figure 3a shows the learned programs. The performance is measured by the average proportion of correct permutations and individual permutations following Grover et al. (2019). Although using weaker background knowledge, $Meta_{Abd}$ has a significantly better performance than NeuralSort in both interpolation (length 5) and extrapolation (length 3 & 7) experiments.

The learned program of `s` and the dyadic neural net `nn_pred` are both successfully re-used in the sorting task, where the learned program of `s` is consulted as interpreted background knowledge (Cropper et al., 2020), and the neural network that generates probabilistic facts of `nn_pred` is directly re-used and continuously trained during the learning of sorting. This experiment also demonstrates $Meta_{Abd}$'s ability of learning recursive logic programs and predicate invention (the invented predicate `s_1` in Figure 3a).

## 5 CONCLUSION

In this paper, we present the Abductive Meta-Interpretive Learning ($Meta_{Abd}$) approach that can train neural networks and learn recursive first-order logic theories with predicate invention simultaneously. By combining symbolic learning with neural networks, $Meta_{Abd}$ can learn human-interpretable models directly from raw-data, and the learned neural models and logic theories can be directly re-used in subsequent learning tasks. $Meta_{Abd}$ is a general framework for combining sub-symbolic perception with logical induction and abduction. The perception model extracts probabilistic facts from sub-symbolic data; the logical induction searches for first-order abductive theories in a relatively small hypothesis space; the logical abduction uses the abductive theory to prune the vast search space of the truth values of the probabilistic facts. The three parts are optimised together in a well-defined probabilistic model.

In future work, we would like to apply $Meta_{Abd}$ on more complicated tasks that involve sub-symbolic perception and symbolic induction, such as reinforcement learning. Instead of approximating logical inference with continuous and differentiable functions, $Meta_{Abd}$ uses pure logical inference for reasoning and it is possible to leverage more advanced symbolic inference/optimisation techniques like Satisfiability Modulo Theories (SMT) (Barrett & Tinelli, 2018) and Answer Set Programming (ASP) (Lifschitz, 2019), which are able to perform large scale inference efficiently.

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
