# OpenReview forum: "Abductive Knowledge Induction from Raw Data"
_ICLR.cc/2021/Conference — Reject_

### Official Review · AnonReviewer3 · 2020-10-26
**Interesting EM framework for bridging perception and reasoning, but with exposition gaps.**

**Rating:** 5
**Confidence:** 4

**Review:**

The paper proposes an EM framework (Meta_abd) for iteratively learning the parameters of a neural network ("perception" component), and inducing the logical rules underlying a domain ("reasoning" component). The rule-learning component is the existing MIL system (with some modifications that are bolded in Figure 2). The paper's ability to learn recursive theories and invent predicates is directly inherited from MIL. Hence, I find its contributions with respect to these two aspects to be incremental at best.

Though interesting, I find that significant details about the Meta_abd's EM model are omitted, and this prevents one from ascertaining Meta_abd's true effectiveness (please see questions below).

Further, EM as a framework for bridging noisy inputs and symbolic reasoning has been explored by prior work (the pLogic, ExpressGNN, and pGAT systems listed below). Could the authors position Meta_abd vis-a-vis these systems, and mention them as related work? Meta_abd's "neural probabilistic facts" map to uncertain knowledge graph triples in these systems (the "ground logical expressions" mentioned in Section 3.1 of the paper), and Meta_abd's neural network's parameters map to entities' embeddings. Like Meta-abd's logical component, the logical part of these systems are used to infer the probabilities of the triples (neural probabilistic facts), which in turn are used to train an embedding model. The difference between these prior systems and Meta_abd is that Meta_abd learns its logical rules whereas the previous systems require the rules to be pre-specified. In the context of these existing systems, the contribution of Meta_abd appears to lie only in learning rules with the existing MIL system, and thus seems moderately incremental.


* Probabilistic Logic Neural Networks for Reasoning. Qu and Tang, 2019
* Efficient Probabilistic Logic Reasoning with Graph Neural Networks, Zhang et al., 2020
* Probabilistic Logic Graph Attention Networks for Reasoning, Vardhan et al., 2020



QUESTIONS

* Page 4, Figure 1: When optimizing the \theta parameters of the neural network in the M-step, are the labels for each digit image probabilistic? If they are, how are the probabilities derived (e.g., from conflicting pseudo-labels)? If the label per image is deterministic, how is the label chosen?

* Page 5, Eqn 5, step 1: How is H *sampled*? From the paper's description, it seems that H is learned deterministically from themodified MIL (Figure 2). Do you perturb the H returned by MIL probabilistically in some way? If not, how does one justify the claim that H is sampled? (Is H the mode?)

* Page 5, Eqn 5, step 2: How much does the the pruning of z with the abduced constraints help? There is no ablation study to verify the contribution of the abduced constraints.

* Page 5, Eqn 5: How are the parameters and/or rules initialized? EM is particularly sensitive to initialization conditions. This is something that the paper alludes to too on page 7 paragraph 2 ("converges to saddle points or local optima"). How does Meta_abd ameliorate this sensitivity? Does it make multiple runs of EM, each from a different initialization? If so, how does it choose the best model? What is the "success rate" of these runs, or how many runs are needed to learn a good model? Are the comparison systems treated fairly in being allowed multiple runs too? The answers to these questions would help to elucidate the workings of Meta_abd.

* Page 6, para 2, "provides a good initialization ... improves the learning performance".
How much does the pre-trained convent help exactly? And from what base?

* Could the authors comment on Meta_abd's scalability? What're its space and time complexity? This information is particularly helpful in view that the empirical evaluation only uses fairly small datasets from a narrow domain.

* Supplementary material, Page 4, Figure 10:
How sensitive is Meta_abd to the meta-rules? Would it be fair to say that the meta-rules are cleverly specified to provide a strong enough structural bias for Meta_abd to be able to learn the correct rules? What happens empirically if the meta-rules are removed (one at a time)? These meta-rules seem to imbue Meta_abd with an unfair advantage over the comparison systems.

---

> ### Author Response · Authors · 2020-11-15
> **Reply to Reviewer 3 (Part 1)**
>
> **Q1:** EM as a framework for bridging noisy inputs and symbolic reasoning has
> been explored by prior work.
>
> **A1:** We thank the reviewer for pointing out these papers, we will add them in
> the related work section in the revised version.
>
>
> **Q2:** In the context of these existing systems, the contribution of Meta_abd
> appears to lie only in learning rules with the existing MIL system, and thus
> seems moderately incremental.
>
> **A2:** We argue that rule learning in neural-symbolic learning is a crucial
> problem, and $Meta_{Abd}$ is not just an incremental work comparing to existing
> approaches.
>
> Firstly, almost all existing method bridging neural net and symbolic reasoning
> are assuming that the given domain knowledge is sound and complete, i.e., once
> the sub-symbolic model like NN can be trained to give correct outputs
> (embeddings, labels, etc.), the reasoning part will always succeed. However,
> this setting enforces users to provide strong domain knowledge before learning,
> which could be impossible in many cases like those of expert systems many years
> ago.
>
> Secondly, first-order logic rules are a kind of programs having nice theoretical
> guarantees (soundness and completeness). A lot of problems can be naturally
> expressed by programs rather than differentiable neural nets, for example the
> sum/product and sorting algorithms in our experiments. Moreover, programs can be
> directly re-used, and they can easily extrapolate to unseen data. The benefits
> have been reflected in our experiments.
>
> Lastly, learning (logic) programs and perceptual neural net together is a
> non-trivial problem. As we have stated in section 3 and previous answers to the
> other reviewers, the interface between sub-symbolic learning and symbolic
> learning is an exponentially-growing set of grounded atoms, only adding MIL
> cannot solve this problem. Previous works in this topic (Dai, et al, 2019;
> Evans, et al., 2019; Li et al., 2020) has shown the difficulty. The major
> contribution of this work is combining abduction and induction to prune the
> search space of the truth-values of the grounded atoms in the interface to make
> the jointly training possible.
>
> **Q3:** Page 4, Figure 1: When optimizing the \theta parameters of the neural
> network in the M-step, are the labels for each digit image probabilistic?
>
> **A3:** The pseudo-labels ($z$) for training the neural net are not
> probabilistic. They are the expectation of the conditional distribution
> $P(z|B,x,y,\theta)$, i.e., the most probable label given background knowledge
> $B$, images $x$, targets $y$ and the current neural network with parameter
> $\theta$. In other words, it is the expectation of the hidden variable $z$ in Eq
> 1 and 2.
>
> **Q4:** How is H sampled?
>
> **A4:** $H$ is sampled during the MIL process. MIL tries to prove each example
> by synthesise a logic program. The synthesise process can be seen as a search
> tree, which can be easily sampled according to the prior distribution.
>
> **Q5:** How much does the pruning of z with the abduced constraints help?
> There is no ablation study to verify the contribution of the abduced
> constraints.
>
> **A5:** There is an ablation study, whose result has been shown in Figure 3b.
> $H\rightarrow z$ is $Meta_{Abd}$, which induces abductive theories $H$ first and
> then use it to constrain the abduction of $z$; $z\rightarrow H$ first performs a
> greedy search on $z$ with probability $p_\theta(z|x)$ to find the most probable
> $z$ and then use it as examples of conventional MIL to learn a valid hypothesis $H$.
> Figure 3b is the average time cost on *one* training example, as we can see, the
> benefit from abduction is huge. We will make it clearer in the revised version.

---

> > ### Comment · AnonReviewer3 · 2020-11-21
> > **Follow-up Questions**
> >
> > Q5/A5: There is an ablation study, whose result has been shown in Figure 3b.
> >
> > Figure 3b shows the impact on the number of prolog inferences. What's the impact on Acc/MAE?

---

> > > ### Author Response · Authors · 2020-11-22
> > > **Follow-up Answers**
> > >
> > > **Q5+**: Figure 3b shows the impact on the number of prolog inferences. What's the
> > > impact on Acc/MAE?
> > >
> > > **A5+**: The Acc/MAE are the same because of the results of logical inferences are
> > > the same. The logical inferences of using/without using abduction are
> > > semantically identical, because they both follow the resolution-based logic
> > > program inference. Since the background knowledge for Meta_abd's
> > > abduction-induction and Metagol's induction are the same, the output programs
> > > and pseudo-labels for updating the neural net are also the same, which does not
> > > affect the Acc/MAE performance.

---

> ### Author Response · Authors · 2020-11-15
> **Reply to Reviewer 3 (Part 2)**
>
> **Q6:** How are the parameters and/or rules initialized?
>
> **A6:** There is no parameter of rules, and the rule structures are learned from
> scratch. Involving probabilities in logic reasoning makes the inference and
> rule structure learning infeasible (see Statistical Relational Learning and
> Probabilistic Graphical Models). This is also the reason why we propose
> $Meta_{Abd}$ and integrate abduction and induction in **pure first-order logic
> inference**. Noises only exist in the raw inputs, which forms the possible
> worlds represented by the probabilistic facts output from neural nets. Please
> see our answer A3 to reviewer 3.
>
> **Q7:** How much does the pre-trained convent help exactly? And from what base?
>
> **A7:** The improvements have been shown in Table 2 and Figure 5 (in appendix).
> The one-shot pre-train of covnet provides a good initialisation of $\theta$,
> which makes EM converges faster and less-probable to be trapped in local
> optima.
>
> **Q8:** The empirical evaluation only uses fairly small datasets from a narrow
> domain. Concerns on Meta_abd's scalability.
>
> **A8:** One of the motivations of introducing relational domain knowledge in
> machine learning is to **reduce the number of required labelled data**.
> Neural-symbolic learning, self-supervised learning and many other machine
> learning areas are aiming at this target. So we think small dataset is not a
> drawback, but a benefit.
>
> The complexity of the induction part of $Meta_{abd}$ is identical to MIL; the
> complexity of abduction is different in each different tasks, because it is
> related to the number of pseudo-labels (potential symbols in raw data).
>
> By enabling full-featured ILP in neural symbolic learning, $Meta_{Abd}$ is able
> to learn large scale knowledge base by simply re-using the learned programs as
> background knowledge in a curriculum of tasks, which is the major benefit of
> symbolic machine learning.
>
> The main target of symbolic machine learning (ILP and program synthesis) is not
> learning "large" programs. Without enough background knowledge, it is impossible
> to learn a good theory/program, because the search space of program grows
> exponential with the length of programs. For example, it is very hard to teach a
> pupil calculus, but it is very easy to teach it to a high-school student.  To
> learn large programs, the most natural way is using curriculum learning, in
> which the background knowledge is continuously accumulated and reused. It can be
> easily realised with $Meta_{Abd}$ because it learns reusable logic theories,
> which is verified by the experiment on learning bogosort.
>
>
> **Q9:** Would it be fair to say that the meta-rules are cleverly specified to
> provide a strong enough structural bias for Meta_abd to be able to learn the
> correct rules?
>
> **A9:** Metarules can be seen as basic grammars in programming language. When
> writing/synthesising a program, the grammars/metarules of course will provide a
> structural bias.
>
> However, we argue that the bias of metarules in this work is not strong at all.
> We use a general set of dyadic metarules, which is not specially picked for any
> task. Removing some of them will even increase the learning speed as it reduces
> the search space.
>
> Comparing to the RNNs, which enforce the end-to-end model to learn a recursive
> program, metarules actually are more general and does not have advantage because
> the resulted space of structural hypotheses is larger and more general in the
> sense of expressive power.

---

> > ### Comment · AnonReviewer3 · 2020-11-21
> > **Follow-up Questions**
> >
> > Thanks for your responses.
> >
> > Q6/A6: "There is no parameter of rules,"
> >
> > I think there may be a misunderstanding. I'm not asking about the parameters of rules, but rather, about how EM is initialized. Modulo convnet, how does Meta_abd initialize EM, and how sensitive are its results to the initialization? (My original comments have more related questions in the same paragraph.)
> >
> > Q9/A9: "we argue that the bias of metarules in this work is not strong at all."
> >
> > Could you quantify the contribution, say, by removing a metarule one at a time and examining the impact on runtime and accuracy/MAE?

---

> > > ### Author Response · Authors · 2020-11-22
> > > **Follow-up Answers**
> > >
> > > **Q6+:** How does Meta_abd initialize EM, and how sensitive are its results to
> > > the initialization?
> > >
> > > **A6+:** Sorry for my misunderstanding.
> > >
> > > The EM is initialized as follows:
> > >
> > > - In $Meta_{abd}$, the perceptual CNN is initialised randomly;
> > > - In $Meta_{abd+\text{1-shot CNN}}$, we randomly sample one example from 0-9 to
> > >   pre-train the CNN as initialisation.
> > >
> > > For the logic rules $H$, $Meta_{abd}$ does not initialise them. They are treated as
> > > hidden variables together with the pseudo-labels $z$ (of the images). However,
> > > the randomly initialised neural net parameter $\theta_0$ results in a randomly
> > > initialised distribution of the probabilistic facts, which will affect
> > > abduction score (Eq. 5) with $P(z|x,\theta_0)$. Therefore, the estimated
> > > the expectation of programs $H$ in the first epoch varies accordingly. For example,
> > > it may output the following hypotheses during the first round of EM in the
> > > accumulative sum experiments:
> > >
> > > ```prolog
> > > f(A,B):-head(A,B).
> > >
> > > f(A,B):-tail(A,C),f(C,B).
> > > f(A,B):-eq(C,B).
> > >
> > > f(A,B):-add(A,C),f(C,B).
> > > f(A,B):-eq(A,B).
> > >
> > > f(A,B):-tail(A,C),f(C,B).
> > > f(A,B):-mult(A,C),eq(C,B).
> > >
> > > f(A,B):-f_1(A,C),f(C,B).
> > > f_1(A,B):-add(A,C),mult(C,B).
> > > f(A,B):-eq(A,B).
> > >
> > > f(A,B):-f_1(A,B).
> > > f_1(A,B):-f_2(A,C),eq(C,B).
> > > f_2(A,B):-mult(A,C),f_2(C,B).
> > > f_2(A,B):-mult(A,C),add(C,B).
> > >
> > > ...
> > > ```
> > >
> > >
> > >
> > > **Q9+:** Could you quantify the contribution, say, by removing a metarule one at
> > > a time and examining the impact on runtime and accuracy/MAE?
> > >
> > > **A9+:** Thanks for your suggestion. As we answered in A5+, the accuracy/MAE
> > > won't change when the hypothesis space defined by metarules contains the correct
> > > programs (there could be multiple semantically identical programs). Otherwise,
> > > the incomplete hypothesis space makes $Meta_{abd}$ fail to estimate the
> > > expectation of $H\cup z$, which causes EM to fail to converge in 100 epochs.
> > >
> > > Following are the time difference measured by the average number of Prolog
> > > inferences in each batch of $Meta_{abd}$'s abduction-induction inference in the
> > > accumulative sum task. The settings are as follows:
> > >
> > > - $Meta_{abd}$ contains at least one metarule, which is `P(A,B):-Q(A,B)`, i.e.,
> > >   calling a primitive function, it is counted as `metarule 1`;
> > > - The perceptual CNN is randomly initialised and un-trained, i.e., the
> > >   distribution of probabilistic facts is random, which is the **worst-case** for
> > >   abduction, so the result here is slower than the average result in Fig. 3b;
> > > - Choosing metarules is a subset selection problem. Following the traditions in
> > >   combinatorial optimisation, we report the **worst result** among all varied
> > >   combinations;
> > > - The number of Prolog inferences includes the CLP(Z) optimisation.
> > >
> > > Results:
> > > - 9 metarules: 26324856 inferences in 1.571 seconds.
> > > - 8 metarules: 26324638 inferences in 1.567 seconds.
> > > - 7 metarules: 26324626 inferences in 1.567 seconds.
> > > - 6 metarules: 26324287 inferences in 1.527 seconds.
> > > - 5 metarules: 26324009 inferences in 1.528 seconds.
> > > - 4 metarules: 26321479 inferences in 1.521 seconds.
> > > - 3 metarules: 26314047 inferences in 1.521 seconds.
> > > - 2 metarules (under this setting, $Meta_{abd}$ is equivalent to RNNs which are
> > >   forced to learn a minimum recursive program): 10991735 inferences in 0.635 seconds.
> > >
> > > As we can see, there is not much difference from 9-3 metarules (when the program
> > > hypothesis space is complete). However, if the users have a strong bias on the
> > > target theory and only use the relevant metarules, the search speed can be
> > > significantly improved.

---

### Official Review · AnonReviewer1 · 2020-10-28

**Rating:** 3
**Confidence:** 4

**Review:**

In this paper, the author proposes Meta_abd which is a hybrid model that learns a deep recognition model and FOL rules the same time. The goal of this work is to learn FOL rules from raw data such as digits presented in image patches in an end-to-end fashion. The model is evaluated with 3 induction benchmarks associated to the MINST digit dataset.

Personally, I find this paper to be difficult to read and many details in methods and experiments are missing, making it hard to understand the authors' contribution. Given its current state, I would recommend rejection. My concerns are as follows:

Motivation:

- Neural-symbolic integration usually refers to combining logic reasoning into deep model's decision process for better interpretability or sample-efficiency, or to use deep models to help the logic reasoning tasks such as ILP or deduction.

- With that being said, I find the claim for this hybrid model to be unjustified. In Meta_abd, NN is only used for data pre-processing which is completely agnostic to the later logic component.

- In fact, diff-ILP (Evans & Grefenstette, 2018) also uses NN for pre-processing MNIST digits for ILP task which I personally find to be similar to the proposed method, though I'm happy to be proven wrong.

Claims: I find many of the claims in the paper to be ambiguous and lack justifications

- In section 2, the author claims that differentiable ILP methods rely on fully trained NN for pre-processing -  this is untrue, for example NeuralLP is an end2end model that can be extended to the MINST benchmark with a perception module that's jointly trainable

- The author also claims that most existing NeSy systems only utilize a pre-defined knowledge base. I find this claim to be confusing and the author does not discuss how the proposed method has addressed this limitation


Method: I find the method to miss many details

- The author defines the learning problem with Eq1 and Eq2 in Section 3.1 and claims to it would be learned through EM. However, Eq1 and 2 do not reveal details about the method, and the exact procedure of EM is unclear to me

- In section 3.3 the author claims the proposed method is an extension of MIL, but this concept is not formally introduced in the paper.

- I find Figure 2 to be difficult to understand - why is Prolog used here? What's the connection of Prolog to the proposed method? It seems to suggest the proposed method is using Prolog for solving the constraints?

Experiment:

- The proposed method is only compared to LSTM and RNN.  The author should include some ILP baselines such as diff-ILP or NeuralLP and substitute the digit image into its ground-truth digit symbol for reference, though I personally think adding a NN pre-processing module is straightforward  as well.

- 3 benchmarks are fairly small consisting of only a few predicates and rules. How does the proposed method scale with the number of predicates and the size of the grounding space?

---

> ### Author Response · Authors · 2020-11-15
> **Reply to Reviewer 1**
>
> **Q1:** In Meta_abd, NN is only used for data pre-processing which is
> completely agnostic to the later logic component. diff-ILP (Evans &
> Grefenstette, 2018) also uses NN for pre-processing MNIST digits for ILP task
> and it is similar to the proposed method.
>
> **A1:** This is not true. In $Meta_{Abd}$, NN is learned simultaneously with
> logic programs. The two components affect each other during the learning
> process, where the logical learning part helps estimate the label of the raw input
> data; the estimated labels are then used to train the neural net to improve its
> classification accuracy; the probabilistic distribution of NN's output labels
> help the logical learning to induce the most probable program.
>
> Partial-ILP is designed for learning symbolic model with gradient-based deep
> learning, which requires symbolic inputs and does not train perceptual networks.
> Please see our answer A3 to reviewer 2.
>
> **Q2:** In section 2, the author claims that differentiable ILP methods rely on
> fully trained NN for pre-processing - this is untrue.
>
> **A2:** Please note that, in section 2, we only say "original ILP, early works on
> combining abduction and induction, and partial ILP are designed for learning
> from symbolic domains (sorry for the typo)". For domains with raw inputs, they
> need to use a fully trained neural model to extract primitive facts from raw
> data before symbolic learning. The motivation of this work is to learn programs
> from raw data.
>
> Most existing works integrates neural nets and logic reasoning do
> not need pre-processing, but they also cannot learn first-order logic rule
> structure and programs. On the other hand, works that can learn logic rule
> structure and programs usually require symbolic inputs (S Russell, Unifying
> Logic and Probability, Comm of ACM, 2015). Hence, they need a pre-processing
> neural network to extract symbols from raw data.
>
> **Q3:** The author also claims that most existing NeSy systems only utilize a
> pre-defined knowledge base. I find this claim to be confusing...
>
> **A3:** We are sorry that we haven't made it clearer about this in section 2.
> Our claim in the abstract and section 1 has stated it in a better way: "they
> (existing NeSy and StarAI methods) usually focus on learning the neural model
> with a **sound and complete symbolic knowledge base**". Our main contribution in
> this work is to enable the learning of recursive logic programs with general
> predicate invention from a weak background knowledge base in a sub-symbolic
> domain, which, to the best of our knowledge, has not been done before.
>
> **Q4:** The author defines the learning problem with Eq1 and Eq2 in Section 3.1
> and claims to it would be learned through EM...and the exact procedure of EM is
> unclear to me.
>
> **A4:** EM is used for parameter learning in problems having hidden variables.
> In Eq1 and Eq2 the $H$ and $z$ in the expectation terms are hidden variables.
> Therefore, EM can solve it by estimating the expectation of $H$ and $z$ and then
> use them to optimise the parameter $\theta$. We will make it clearer in the
> revised version.
>
> **Q5:** Introduction to MIL and relation to Prolog.
>
> **A5:** Meta-Interpretive Learning is an Inductive Logic Programming framework
> implemented with Prolog language. We will add more text to introduce them in the
> revised paper.
>
> **Q6:** The author should include some ILP baselines such as diff-ILP or
> NeuralLP and substitute the digit image into its ground-truth digit symbol for
> reference.
>
> **A6:** As claimed in section 1 and 2, pre-training and pre-processing are what
> we try to avoid in this work. The motivation of this work is to combine
> sub-symbolic and symbolic learning, in which the labels for training the
> perception model are **unknown**. Therefore they should be trained jointly. If
> we use pre-trained NN to pre-process raw data into symbols, we could directly
> apply StarAI or ILP to learn symbolic models. However, in this way, we need to
> provide the label for training the NN model, which could be impossible in many
> applications.
>
> **Q7:** 3 benchmarks are fairly small consisting of only a few predicates and
> rules. How does the proposed method scale with the number of predicates and the
> size of the grounding space?
>
> **A7:** We argue that the three benchmarks are not small in the area program
> synthesis and inductive logic programming. The learned programs are recursive;
> one of them even requires predicate invention. These tasks are all non-trivial
> in the area of symbolic machine learning. Moreover, the main challenge lies the
> uncertainty of "what symbols exist in the raw data". For the ILP model, because
> the input facts extracted from raw data are uncertain, the program induction
> becomes even harder. Scalability of ILP and program synthesis w.r.t. the size of
> background knowledge remains an open problem in these areas, which is not the
> main issue of our paper.

---

### Official Review · AnonReviewer2 · 2020-10-28

**Rating:** 4
**Confidence:** 5

**Review:**

Summary.

In this paper, the authors have presented a framework that combines meta-interpretive learning and the abductive learning of neural networks. The high-level idea is to formulate a unified probabilistic interpretation of the entire algorithm so that both the inductive logic programming module and the neural network modules can be trained jointly from data. The authors have demonstrated the application of the proposed algorithm to learning arithmetic operations and sorting operations by looking at input-output mnist digits.

Comments.
The key idea of the paper has been presented clearly. The authors demonstrated two tasks: cumulative sum/product, and sorting. Both tasks require learning recursive rules, and the bogosort task requires predicate invention. These are challenging tasks for both neural networks and ILP algorithms.

However, my major comments about the paper is that the experiment sections are relatively weak and they have definitely missed some important baseline comparisons. Concretely, taking the cumulative summation task as an example, the MetaAbd model has very strong inductive biases, because of the builtin "add" operation and the metarules built into the system, which strongly favors recursive rules of specific forms. However, at least the "add" operation was not built into other baselines.

Second, there have also been many other works trying to solve this task:
- partial ILP (Evans & Grefenstette, 2018) and machine apperception (Evans et al.,
2019) that can learn mnist digits with much weaker assumptions: they can even learn the "succ" relationship between digits.
- Neural GPU (Kaiser and Sutskever 2015) that can learn to add multi-digit numbers without any builtin "add" operations.
- Differentiable Neural Computer (https://deepmind.com/blog/article/differentiable-neural-computers)
- Neural Programmer-Interpreters (Reed et al 2015) and its follow-ups: they support integrating human-written primitive functions (such as the "add" operation) with neural networks.
The authors are encouraged to make comparisons with these methods as well.

Third, the learned logic rules are relatively simple. This makes me less convinced about the applicability of the paper. The authors have made very strong claims in the abstract/intro about "To the best of our knowledge, MetaAbd is the first system that can jointly learn neural networks and recursive first-order logic theories with predicate invention." For example, partial ILP and machine apperception can do that, too. Recently, there have also been other trials on using relational neural networks for bridging perception and rule learning, such as,
- Graph Neural Networks (https://arxiv.org/abs/1806.01261)
- Neural Logic Machines (https://arxiv.org/abs/1904.11694)

Overall, I think this paper is not matching the publication standard of ICLR.

Minor:
Please change the latex formatting of the model name. There is currently an extra space between M and e.

---

> ### Author Response · Authors · 2020-11-15
> **Reply to Reviewer 2**
>
> **Q1:** The MetaAbd model has very strong inductive biases, because of the
> builtin "add" operation and the metarules built into the system, which strongly
> favors recursive rules of specific forms.
>
> **A1:** We argue that, comparing to the compared end-to-end methods, the
> background knowledge exploited by $Meta_{Abd}$ is not strong at all.
>
> The NAC and NALU module has built-in functions of summation and multiplication;
> the RNNs force the end-to-end model to learn recursive function. Meanwhile,
> $Meta_{Abd}$ also uses summation and multiplication as primitives; the metarules
> are set of generic metarules (shown in figure 10 in the appendix) which do not
> have any preference on learning recursive theories. Table 1 briefly summarised
> the comparison of domain knowledge between $Meta_{Abd}$ and the compared
> methods; we will add more clarifications in the revised version.
>
>
> **Q2:** The experiment sections are relatively weak and they have definitely
> missed some important baseline comparisons.
>
> **A2:** Although the tasks are made from synthetic examples, they are not
> trivial problems for both end-to-end models and neuro-symbolic methods. As we
> can observe from the experimental results, even though all the compared methods
> have used a certain amount of domain knowledge, the problems are still hard to
> solve. For more explanations, please refer to our answer A7 to reviewer 1.
>
> We appreciate that the reviewer points out more related works, we will add more
> discussion (please see A3-A5) about them in the revised version.
>
>
> **Q3:** partial ILP (Evans & Grefenstette, 2018) and machine apperception (Evans et
> al., 2019) that can learn mnist digits with much weaker assumptions: they can
> even learn the "succ" relationship between digits.
>
> **A3:** The input to Partial ILP are symbolic data, which requires a pre-trained
> CNN to extract the raw data into symbols; Machine Apperception uses a Binary
> Neural Net to learn perception; however, it cannot handle very noisy inputs due
> to the high complexity of BNN that is optimised by answer set programming (a
> discrete search-based approach like SAT solvers). In fact, it is possible to use MNIST
> images for the Seek Whence task. However, due to the highly noisy inputs, the ASP-based
> optimisation of Binary Neural Network for image recognition is very inefficient.
> That's why we say it is difficult for partial ILP and Machine Apperception to
> jointly learn NN and logical theories from noisy sub-symbolic domains like MNIST
> images.
>
> Specifically, Machine Apperception can only use Binary Neural Net for
> perception, while $Meta_{Abd}$ is a more general approach that does not
> constrain the type of perception model. Furthermore, during the Machine
> Apperception engine can invent concepts of objects (i.e., monadic predicate),
> $Meta_Abd$ can perform general predicate invention, which subsumes object
> invention. We are sorry that we haven't made it clearer in the submission and
> will revise it in the next version.
>
> $Meta_Abd$ has successfully learned the "success" relation between images in the
> bogosort task, even without background knowledge about that success is a
> transitive relation. More details are described in the appendix A.3, Paragraph
> Example of Dyadic facts abduction, the dyadic neural predicate `nn_pred/2`
> learns the success relation automatically from the MNIST images.
>
>
> **Q4:** Neural GPU (Kaiser and Sutskever 2015), Differentiable Neural Computer,
> Neural Programmer-Interpreters (Reed et al 2015), Neural Logical Machines.
>
> **A4:** All these methods are assuming symbolic inputs, which means they are not
> designed for learning perception and reasoning. For example, in (Dai et al.,
> Bridging machine learning and logical reasoning by abductive learning, NeurIPS
> 2019), it has been shown that Differentiable Neural Computer does not perform
> well in heavy-reasoning tasks with raw inputs.
>
>
> **Q5:** About Graph Neural Networks.
>
> **A5:** As we have discussed in section 1, GNNs usually require a fixed
> relational graph structure before learning, while the motivation of this work is
> to learn such relational structures (e.g., logic programs) when they are not
> available.
>
> **Q6:** The learned logic rules are relatively simple. This makes me less
> convinced about the applicability of the paper.
>
> **A6:** The difficulty of the tasks is not only coming from the rules, but also
> resulted by the noisy raw data inputs. Therefore, we argue that these tasks are
> non-trival. Please see our answer A7 to Reviewer 1.

---

> > ### Comment · AnonReviewer2 · 2020-11-24
> > **Thanks for the Clarifications**
> >
> > Thank you for the clarifications. My biggest concern about this paper remains to be missing comparison with other related works.
> >
> > In general, combining rule learning and pattern recognition tools, such as deep networks have been broadly studied. The authors have presented a hybrid approach of deep nets + ILP. They are connected by an abductive reasoning layer.
> > However, there have been other approaches in this field.
> > - Partial ILP and Machine Apperception use gradient-based methods to connect deep nets learning and ILP. For Partial ILP, if you treat the inference engine as a black box, it naturally provides gradients to its input, and thus the entire system can be optimized with gradient descent.
> > - Neural GPU, Neural logic machines use end-to-end neural architectures to simulate forward chaining. They can realize logic rules of certain forms.
> > - Differentiable neural computer and neural programmer-interpreters use policy gradient to differentiate through non-differentiable primitive operations.
> >
> > Although many of these works have been originally tested on symbolic inputs only, or used pretrained neural networks, they can definitely be extended to image inputs. I still think that the current experimental setups do not fully demonstrate the power of the proposed methods.
> >
> > Given all that, I am not willing to increase my rating to a positive one at this moment.

---

### Official Review · AnonReviewer4 · 2020-10-28
**Potentially exciting, but not explained well enough**

**Rating:** 4
**Confidence:** 4

**Review:**

I like the idea of this paper, however the paper seems to be more written to impress rather than inform.

I cannot see how it "learns ... simultaneously from raw data."  I think the interface is in the nn(image=value,prob) but nn() is never explained in the text. I can't work out: what are the inputs? What are the outputs? What. is the relationship between them? Similarly,  prove() used in Figure 2 is never explained; what are the argument? What is the intended interpretation?

It seems funny to have 0/1 probabilities (equation (4)). Surely there is so much noise that you can't perfectly predict the outputs.

I didn't see how you overcome the exponential complexity promised in the abstract (and why do you think it is caused by the interface?).

In the experiments, why is MAE (or log MAE) a reasonable measure? What is the accuracy of the correct program? (I think there is supposed to be a correct program you are learning).  Is the correct program in the search space with a non-zero prior?

Is "Acc" in table 2 correspond to just predicting the digit? Why is there so much variability? What is the accuracy of nn()?

The paper needs to be self contained. For example, you need to tell us that #= means equality and what Prolog's permutation  predicate is  (is it related to permute() in Figure 3?)

I understand you are trying to learn the simplest logic program that can produce the output from raw images. What you did at the top-level seems right, but it is not described well enough. You need to provide enough details so that it is reproducible.

---

> ### Author Response · Authors · 2020-11-15
> **Reply to Reviewer 4**
>
> **Q1:** `nn()` is never explained in the text.
>
> **A1:** We are sorry that we haven't clarified the meaning of `nn` predicate in
> the main text, it is explained in the appendix A.2 (second paragraph on page
> 12). We will revise to make it clearer.
>
> **Q2:** what are the inputs and outputs? What is the relationship
> between them? prove() used in Figure 2 is never explained.
>
> **A2:** The inputs and outputs and their relationship have been clearly
> explained in the first and second paragraph of section 3.1.
>
> Figure 2 is a Prolog meta-interpreter, which is briefly explained in the second
> paragraph of section 3.3. Considering that most of audience of ICLR does not
> have a background in ILP or program synthesis, we will add detailed explanatory
> text for the algorithm in the revised version.
>
>
> **Q3:** It seems funny to have 0/1 probabilities (equation (4)). Surely there is
> so much noise that you can't perfectly predict the outputs.
>
> **A3:** First-order logical reasoning **should be** noise-free, that's why it
> has solid theoretical guarantees like soundness and completeness.
>
> $Meta_Abd$ adopts the *distribution semantics* in probabilistic logic (T Sato, A
> statistical learning method for logic programs with distribution semantics, ICLP
> 1995), in which the noises are handled by the **possible worlds** of ground
> facts, in each possible world, the inference is deterministic. For example in
> Figure 2, the interpretation of MNIST images are noisy, but once the symbol
> (i.e., digits) are recognised, the logical inference for summation will only
> have 0/1 probabilities, isn't it?
>
>
> **Q4:** why do you think (exponential complexity) is caused by the interface? how
> you overcome it?
>
> **A4:** For example, given 3000 MNIST sequences with length 5, there are 15000
> labels to be predicted, i.e., $10^15000$ possible pseudo-label ($z$)
> assignments. The pseudo-label $z$ is used to induce a program to calculate the
> final label ($y$, e.g., the sum of each MNIST sequence). Without an accurate
> $z$, it is very difficult to learn the correct program. Hence, the challenge is
> introduced by the interface $z$.
>
> We overcome it by combining abduction and induction. $Meta_{Abd}$ learns an
> abductive logic program $H$, and then uses $H$ to infer (which is called
> abduction in first-order logic) the possible values of $z$. A detailed
> explanation is in the second paragraph of section 3.2; and a more detailed
> explanation is in the appendix (first paragraph on page 12).
>
>
> **Q5:** why use MAE (or log MAE)? What is the accuracy of the correct program?
> Is the correct program in the search space with a non-zero prior?
>
> **A5:** MAE is the evaluation metric in the original paper of NALU, and we
> agree with them because summation and multiplication are regression tasks.
>
> The accuracy of learning programs is either 0 or 1, i.e., whether $Meta_{Abd}$
> outputs a correct program (covers all positive examples and no negative example)
> or a wrong program (does not cover all positive examples or covers at least one
> negative example), following the definition of Inductive Logic Programming, as
> we explained in *A3*. The learned programs are clearly shown in Figure 3a.
>
>
> **Q6:** Is "Acc" in table 2 correspond to just predicting the digit? Why is
> there so much variability? What is the accuracy of nn()?
>
> **A6:** Yes, "Acc" is the accuracy of the learned CNN, which predicts the label
> of a single image. The metric is also used in the original NALU paper. It is the
> final accuracy of `nn`. The training curve of `nn` is shown in Figure 5 in the
> appendix.
>
>
> **Q7:** The paper needs to be self contained. Concerns on reproducibility.
>
> **A7:** We are sorry that we have put too many details in the Appendix, and we
> will improve the presentation by moving the important contents into the main
> text. We have a section about reproducibility in the Appendix, our codes and
> data are also available in the supplementary materials.

---

### Author Response · Authors · 2020-11-15
**About motivation and contribution**

Thanks to all the reviewers for the detailed comments. After reading all the
reviews, we found that most of the reviewers do not have much background in
Inductive Logic Programming (ILP) or program synthesis. We are sorry that our initial
submission hasn't included more preliminaries about these subjects, which will
be added in the revised version.

In order to help the reviewers and readers understand our work better, we mildly
introduce ILP and clarify our main contribution and motivation as follows.

=== Inductive Logic Programming and Meta-Interpretive Learning ===

Inductive Logic Programming is a symbolic machine learning paradigm for learning
first-order logic theories from data. Given some background knowledge $B$ and a
set of examples $E=E^+\cup E^-$, where $E^+$ are positive examples and $E^-$ are
negative examples, the target of ILP is to induce a hypothetical first-order
logic theory $H$, such that $B\cup H\models E+$ and $B\cup H\not\models E^-$.
Because $B$, $E$ and $H$ are all represented by *Logic Program* (e.g., Prolog
program), this machine learning technique is named as *Inductive Logic
Programming*, i.e., "inducing logic programs from data".

Briefly speaking, ILP has the same target of program synthesis, which tries to
teach computers how to write programs by providing a number of working
examples ($E$) and some primitive functions ($B$). But different to ordinary
program synthesis, ILP uses *first-order logic* as its programming language,
which has a firm and solid foundation in mathematics, it brings ILP some very
important theoretical guarantees such as *soundness* and *completeness*. For
more information about ILP, please refer to the very good introductory
[article](https://arxiv.org/pdf/2008.07912.pdf) by Andrew Cropper and Sebastijan
Dumančić recently.

Meta-Interpretive Learning (MIL) is an ILP approach that implements a
second-order logical meta-interpreter for learning first-order logic. The
meta-interpreter tries to prove the positive training examples (and disprove the
negative ones) by backward chaining. During the proving procedure, MIL uses
metarules as templates to construct logic programs as the hypothesis $H$. Once
$H$ can satisfy all positive examples and disprove all negative examples then
the learning process will be terminated, otherwise it will backtrack and search
for another hypothesis. Many neural-symbolic algorithms use a similar idea to MIL and
metarules to learn logic rules, for example, the Neural Theorem Prover
(Rocktaschel et al., 2017) and partial-ILP (Evans and Grefenstette, 2018).

=== Why Integrating Neural Networks with ILP ===

Although end-to-end deep learning has achieved great success in many areas, it
still have some drawbacks.

There are many problems that should be solved by high-level reasoning. Algorithms and
programs are the most important components in computer science, they use formal
language to represent and solve problems. There are many (System 2) tasks in AI
can be naturally expressed by formal language and solved by programming. As a
generic program synthesis method with a solid mathematical foundation, we believe
ILP suits such purpose very well.

Take the MNIST summation task in our paper as an example, the training examples
consist of a sequence of MNIST images as input and an associated number as
output. To learn a model to solve this problem, the most natural idea (at least
for us) is to divide the task into two steps: 1) recognise the digits from
images; 2) learn a program to calculate the digits. Hence, task 1) should be
solved by neural nets, which is very good at learning perceptual model from
sensory data; task 2) can be solved by program synthesis or ILP.

Because ILP is formulated with first-order logic, which is a general
way to express knowledge. The background knowledge, examples and hypotheses are
all represented in a unified way. As a result, humans can easily inject domain
knowledge in ILP-based machine learning, the learned models are interpretable,
*debuggable* and *reusable*.

However, ILP and program synthesis both requires symbolic inputs, i.e., the
examples and background knowledge are logic programs. In real applications,
the data is usually noisy. Therefore, ILP and program synthesis cannot be
directly applied on this kind of tasks, and previous work that tries to induce
logical theory or computer programs from raw data requires a pre-processing step,
which filters the noises and extracts symbols from raw data.

---

> ### Author Response · Authors · 2020-11-15
> **More clarifications**
>
> === Contribution of This Work ===
>
> The main contribution of this work is combining neural network with
> full-featured Inductive Logic Programming and let the two systems to be trained
> simultaneously in a unified framework. To the best of our knowledge, this is
> the first work does this job. Nevertheless, our knowledge is limited, we are
> happy to hear comments about related works and works that sharing similar ideas.
>
> However, simply combining the two systems requires an interface between them.
> The ILP system needs symbolic inputs for logical inference (both induction and
> deduction); the neural net requires labels for supervised training. The
> interface is discrete, and its search space grows exponentially with the number
> of examples. To make the simultaneously training possible, we propose to
> integrate induction and abduction to reduce the cost of the optimisation.
>
> Abduction can be seen as "reversed deduction". Different from induction, it does
> not learn any "general rule" from empirical data. Instead, it tries to explain
> the observed outcome based on background knowledge. For example, when we see the
> grass in front of our house is wet, abduction will tell us it might have rained
> or the sprinkler was on. As an analogy to gradient descent in the deep learning
> context, abduction "propagates the error" backwards from the output facts to
> find the most probable input values.
>
> By integrating abduction and induction, our proposed method can drastically
> reduce the search time (see Figure 3b) of truth values of the unknown interface
> symbols (labels of the images), which makes training neural networks with
> full-featured ILP together possible.
>
> === Baseline Methods and Experimental Tasks ===
>
> As we have mentioned before, integrating neural nets and ILP will: 1) enable
> machine learning the ability to exploit domain knowledge expressed with formal
> language; and 2) learning human interpretable programs from raw data. Therefore,
> the baseline methods of end-to-end deep neural networks should 1) exploit the
> same (or at least equivalent) domain knowledge; 2) is designed for learning
> "program-like" models from sub-symbolic data. Neural Arithmetic Logic Units and
> Neural Sort are the most representative examples in this category. This is also
> why we use the same tasks and performance measures in their original papers.

---

### Decision · Program_Chairs · 2021-01-07
**Final Decision**

**Decision:**

Reject

**Comment:**

The paper addresses the difficult problem of combining ILP in a meta-interpretive framework with noisy inputs from a neural system.   The essential idea is to use MIL to "efficiently" search for constraints on the neural outputs (eg z1 + z2 + z3 = 7, or z2< z3) as well as logic programs, with a score related to program complexity as well as probability of the best constraint-satisfying neural outputs.  It is interesting work for the right audience but it's clear from the reviews that the presentation was difficult for ICLR readers, even ones with appropriate background.

Some potential weaknesses of the approach include:

1 - it's unclear how scalable the MIL framework is - presumably the intrinsic difficultly of the search means that programs and constraint sets must be small

2 - it's unclear how general the approach is beyond the digits-as-separate-inputs setting of the two experimental studies, and its unclear how accurate the perceptual layer needs to be - MNIST obviously being an example of a case where there is little noise with a modern classifier.

3 - it's unclear how constraints can in general be used to backprop any information to the underlying neural system, and without this the joint training seems to be quite limited.

Overall the paper is judged as inappropriate for ICLR.